# Predicting Continuous Chip to Segmented Chip Transition in Orthogonal Cutting of C45E Steel through Damage Modeling

**Ashwin Moris Devotta [1],\*, P. V. Sivaprasad [2] , Tomas Beno [3] and Mahdi Eynian [3]**

1   R&D Turning, Sandvik Coromant AB, 81181 Sandviken, Sweden
2   R&D, Sandvik Materials Technology AB, 81181 Sandviken, Sweden; palla.sivaprasad@sandvik.com
3   Department of Engineering Science, University West, SE-46132 Trollhättan, Sweden; tomas.beno@hv.se (T.B.);
    mahdi.eynian@hv.se (M.E.)
\*   Correspondence: ashwin.devotta@sandvik.com; Tel.: +46-706-163-722

**Abstract:** Machining process modeling has been an active endeavor for more than a century and it has been reported to be able to predict industrially relevant process outcomes. Recent advances in the fundamental understanding of material behavior and material modeling aids in improving the sustainability of industrial machining process. In this work, the flow stress behavior of C45E steel is modeled by modifying the well-known Johnson-Cook model that incorporates the dynamic strain aging (DSA) influence. The modification is based on the Voyiadjis-Abed-Rusinek (VAR) material model approach. The modified JC model provides the possibility for the first time to include DSA influence in chip formation simulations. The transition from continuous to segmented chip for varying rake angle and feed at constant cutting velocity is predicted while using the ductile damage modeling approach with two different fracture initiation strain models (Autenrieth fracture initiation strain model and Karp fracture initiation strain model). The result shows that chip segmentation intensity and frequency is sensitive to fracture initiation strain models. The Autenrieth fracture initiation strain model can predict the transition from continuous to segmented chip qualitatively. The study shows the transition from continuous chip to segmented chip for varying feed rates and rake angles for the first time. The study highlights the need for material testing at strain, strain rate, and temperature prevalent in the machining process for the development of flow stress and fracture models.

**Keywords:** chip segmentation; damage modeling; dynamic strain aging

## 1. Introduction

The machining process remains one of the critical manufacturing processes in the 21st century and it has critical engineering applications [1]. Developments in the fields of plasticity and fracture mechanics are used to improve the understanding of the machining process [2]. This improved understanding has become even more necessary with the ever-increasing reliability need for engineered components [3].

Machining process modeling has been carried out while using different approaches, such as empirical, analytical and numerical methods [4]. The finite element (FE) method has been quite extensively used in the modeling of the machining process to model the chip formation process. FE modeling of the machining process provides the ability to incorporate a newer advanced understanding of material behavior. This improved understanding is usually obtained through other methods, such as material testing or more sophisticated modeling techniques. Within FE modeling of the machining process, workpiece material modeling requires the material response to large deformations (large plastic strains) at very high strain rates and very high temperatures.

The ferritic-pearlitic steel group is one of the essential engineering steels having a wide area of application [5]. Material deformation mechanisms, such as strain hardening, strain rate hardening, and thermal softening are vital and have been used in FE modeling of the machining process for more than two decades. Besides, there are deformation mechanisms that are material specific, such as dynamic strain aging (DSA). Previous studies carried out in understanding the behavior of ferritic-pearlitic steel have shown that DSA is a function of temperature and strain rate [6–9]. Earlier studies have been uncertain about the need for the incorporation of DSA in machining process modeling [10,11]. However, recent studies [9,12] have shown the need to incorporate DSA to improve the model's prediction accuracy. DSA has been shown to influence strain hardening behavior, thermal softening behavior, and strain rate hardening [13]. C45E steel falls within the ferritic-pearlitic dual-phase steel group. Flow stress experiments in C45E steel through compression testing at varying temperatures and strain rates have shown the presence of DSA [14]. A modified form of the Johnson–Cook (JC) model was developed using the regression modeling approach. In another recently reported work [8], a material model (Voyiadjis-Abed-Rusinek (VAR) model) was developed, where a phenomenological model is combined with a physics-based model to capture DSA in C45 steel and it is better suited for FE simulation implementation. Using the VAR approach, an attempt has been made to modify the JC model to incorporate DSA. Further, the modified JC (MJC) model is to be implemented to simulate chip formation in the machining process.

Predicting chip segmentation through FE simulations requires the modification of flow stress curves through strain-softening modifications [15] or damage modeling [16–20]. Direct strain-softening modifications of flow stress are based on the adiabatic shear theory [21]. They have been used primarily in chip segmentation prediction in the machining of Ti alloys. Damage modeling where the material failure due to the ductile shear failure has also been used in the chip segmentation of ductile material machining. Most of the studies have concentrated on improving chip segmentation prediction accuracy in terms of chip segmentation intensity [22]. Besides, from a machining process design perspective, the transition from continuous chip to segmented chip is necessary and it has been the focus of very few studies [22,23]. The continuous chip to segmented chip transition with increasing cutting velocity is attributed to the adiabatic shearing process [24]. Recently developed models [16] have explored this area where the influence of damage parameter on chip segmentation has been evaluated. The ability of finite element simulations to predict the continuous chip to segmented chip transition is scarce [23], and none exists for steel machining to the authors' knowledge.

The present work aims to modify the Johnson–Cook model for the incorporation of the DSA influence. The modified Johnson–Cook model, in combination with the damage model, needs to be implemented then within an FE framework for simulation of chip formation in machining. The fracture initiation strain model's influence within the ductile damage model approach is to be evaluated in the prediction of the continuous chip to segmented chip transition in the machining of C45E steel under orthogonal cutting conditions.

In this section, the development of the modified JC model for C45E steel based on the approach developed for the VAR model [8] is presented.

## 2. Modified Johnson–Cook Model Development Incorporating DSA

### 2.1. Modified Johnson–Cook Model

The JC model, which is one of the most used material models in the finite element simulation of the machining process, as described in Equation (1).The JC model is built in the multiplicative form in the form of strain hardening component, strain rate hardening component and thermal softening component [25]. The strain hardening component is defined by initial yield stress (*A*), strain hardening coefficient (*B*) and strain hardening exponent (*n*), respectively. The strain rate hardening and thermal softening terms are fitted using the parameters *C* and *m*, respectively.

The fitting of the strain hardening component is traditionally carried out using standard quasistatic testing using tensile or compressive loading. In the previously reported work, *A* and *B* were modeled using flow stresses at varying temperatures and strain rates using regression modeling approach. The strain hardening exponent, *n*, was fitted as a first-order function of temperature. The strain hardening behavior of the JC model itself is written in the form, as shown in Equation (2) to reduce the FE implementation complexity.

$$\overline{\sigma} = (A + B\overline{\varepsilon}^n)\left(1 + C\ln\left(\frac{\dot{\varepsilon}}{\dot{\varepsilon}_0}\right)\right)\left(1 - \left(\frac{T - T_o}{T_m - T_0}\right)^m\right) \tag{1}$$

$$A + B\varepsilon^n = A\left(1 + \frac{B}{A}\varepsilon^n\right) \tag{2}$$

$$A = A_0 + A_\Delta\left[1 + \tanh\left(\frac{T - T_l}{\xi}\right)\right]\left[1 - \tanh\left(\frac{T - T_h}{\xi}\right)\right] \tag{3}$$

$$A_\Delta = \Delta\sigma\left(1 + A_1\log\left(\frac{\dot{\varepsilon}}{\dot{\varepsilon}_0}\right)\right) \tag{4}$$

$$T_l = T_{ll}\left(1.15 + t\log\left(\frac{\dot{\varepsilon}}{\dot{\varepsilon}_0}\right)\right) \tag{5}$$

$$T_h = T_{hh}\left(1.15 + t\log\left(\frac{\dot{\varepsilon}}{\dot{\varepsilon}_0}\right)\right) \tag{6}$$

The initial yield stress parameter A is modified similarly to the VAR model to incorporate the DSA influence, leading to yield stress increase with temperature increase at specific temperature ranges [8], as in Equations (3)–(6) and shown in Figure 1. Reported experimental investigations have shown that the dynamic strain aging temperature range is a function of strain rate [6,7,14]. The value of $A_0$ is obtained from the room temperature quasistatic compression test. The value of $A_\Delta$ corresponds to the peak initial yield stress increase in the DSA regime. The constants $A_1$ and $t$ are fitted with flow stress curves that are obtained at high strain rate conditions from El Magd et al. [6] and Hokka et al. [7]. The temperatures $T_l$ and $T_h$ represent the start and end of DSA, the regime at the reference strain rate. $T_l$ and $T_h$ are modeled as a function of strain rate instead of a constant as in the VAR model. This methodology is necessary to accommodate the observation of dynamic strain aging temperature range being dependent on strain rate [6,7]. The model assumes the DSA regime to extend to higher strain rates at which experimental results are unavailable. Nevertheless, the DSA regime for machining process modeling has been incorporated by modifying the temperature component of the Power-law model while using a regression equation by Childs et al. [26], with the limitation of the temperature range to be a constant. The constants, $B/A$ and $n$, are fitted for three different temperature ranges to accommodate the influence of temperature on strain hardening.

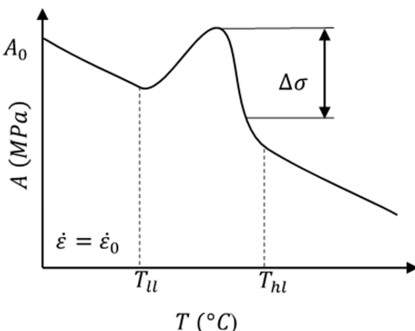

**Figure 1.** Schematic of the Voyiadjis-Abed-Rusinek (VAR) model-based modification of the JC model's initial yield stress parameter with corresponding mathematical equations with A0 representing the constant, A, of the JC model.

## 2.2. Fitting of Flow Stress Curves Using the Modified JC Model

The ability of the newly developed flow stress curves from the previous section to predict DSA has been validated using the Gleeble test data from previous work [14]. The flow stress curves are obtained for three different strain rates of 1 s$^{-1}$, 5 s$^{-1}$, and 60 s$^{-1}$ with programmed temperatures between 200–700 °C. The temperature range at which DSA is exhibited is a function of the strain rate [6,7]. In the temperature-strain rate range where DSA is active, the initial yield stress increases with temperature and decreases with the strain rate. The presence of DSA has also been shown to lead to serrations in the flow stresses. The temperature at which DSA peaks has been identified to be around 650 °C at high strain rates present in machining conditions [12]. Table 1 presents the constants of the modified JC model. The parameter $B$ and $n$ in the modified Johnson-Cook model has been modified for three different strain hardening regimes. This approach is carried out to ease the implementation of the FE method. The constants are provided, as shown in Equation (7).

$$\begin{cases} B = 1.5A, & n = 0.5 & T < 400\,^{\circ}\mathrm{C} \\ B = 1.2A, & n = 0.3 & 400\,^{\circ}\mathrm{C} < T < 500\,^{\circ}\mathrm{C} \\ B = 1.0A, & n = 0.2 & T > 500\,^{\circ}\mathrm{C} \end{cases} \tag{7}$$

**Table 1.** Modified JC model parameters for normalized C45E steel.

| $A_0$ (MPa) | $\Delta\sigma$ (MPa) | $A_1$ (-) | $T_{ll}$ (°C) | $T_{hh}$ (°C) | $t$ (-) | $\xi$ (-) | $C$ (-) | $m$ (-) |
|---|---|---|---|---|---|---|---|---|
| 500.0 | 80.0 | 0.0001 | 200.0 | 500 | 0.1 | 100 | 0.0018 | 1.0 |

The strain rate hardening and thermal softening parameters of the original JC model for C45E steel are used in this study ($C$ = 0.0018 and $m$ = 1) [11]. With the parameters obtained from the MJC model, the flow stress curves at a strain of 0.1 are predicted under different temperatures and three different strain rates are shown, as in Figure 2. The flow stress prediction extrapolated to higher strain rates prevalent in machining conditions is also plotted to visualize the DSA regime being a function of strain rate.

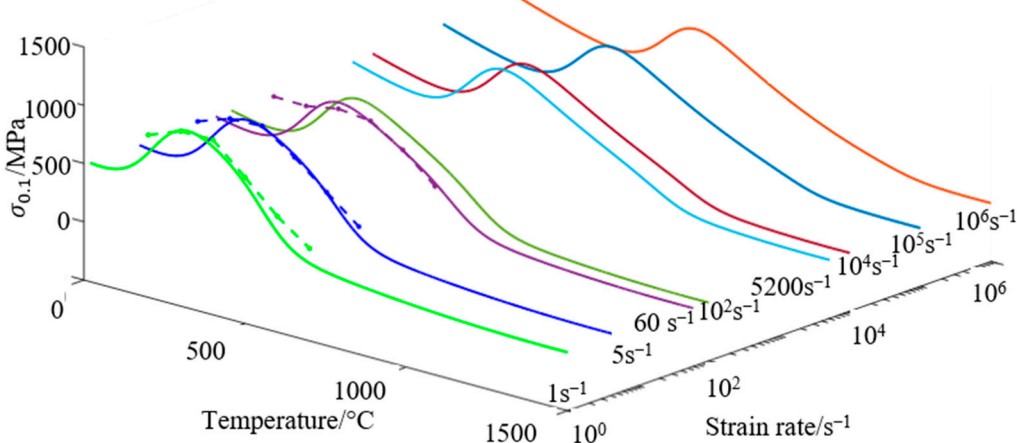

**Figure 2.** Initial yield stress ($\sigma_{0.1}$) for varying temperature and strain rate with the modified Johnson–Cook (JC) model extrapolated to high strain rates to accommodate machining conditions (Experimental data is plotted with dashed lines and model-predicted data is plotted with continuous lines).

The modification of the initial yield stress parameter of the JC model can capture the increase in initial yield stress with temperature increase. At the reference strain rate of 1 s$^{-1}$, the DSA regime is active in the temperature range of 200 °C to 500 °C. With a strain rate increase between 10$^3$ s$^{-1}$ to

$10^6$ s$^{-1}$, the modeled DSA regime's initial yield stress moves to higher temperatures of 650 °C and it correlates with other published experimental results [6,7].

With the ability of the MJC model's initial yield stress parameter to capture the DSA regime in Figure 2, Figure 3 shows that the MJC model can capture the strain hardening behavior observed from Gleeble tests with reasonable accuracy. In the temperature ranges below 400 °C, the material strain hardens with increasing strain. At 500 °C, dynamic recovery influences strain hardening, leading to a lowering slope, and are well captured by the MJC model. Beyond 500 °C, at 600 °C and 700 °C, the flow stress curves exhibit constant flow stress with increasing strain.

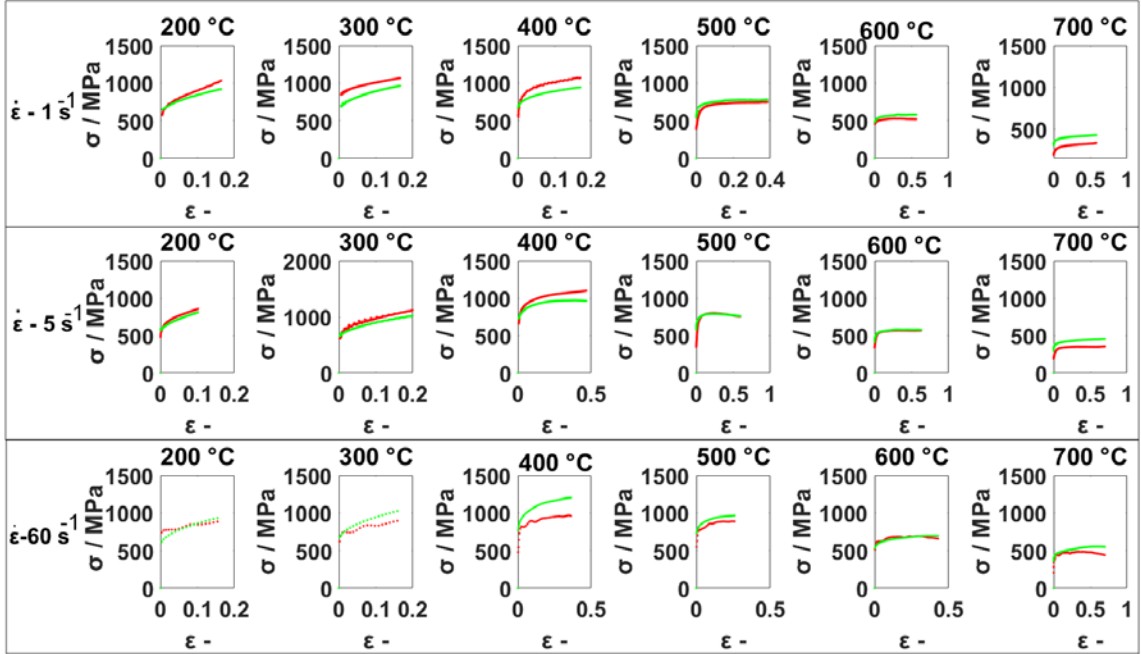

**Figure 3.** Flow stress curves predicted by the modified JC (MJC) model compared with the flow stress curves obtained using compression tests using Gleeble thermomechanical simulator at strain rates 1 s$^{-1}$, 5 s$^{-1}$ and 60 s$^{-1}$ (Model: Green color; Experimental: Red color).

MJC model predicted $\sigma_{0.1}$ at a $\dot{\varepsilon}$ of $0.52 \times 10^4$ s$^{-1}$ is plotted in Figure 4 to evaluate the validity of extending the hypothesis of DSA presence at very high strain rates. The model can predict $\sigma_{0.1}$ with reasonable accuracy. The flow stress at a strain of 0.1 is chosen to avoid the transient conditions during the early stages of loading in compression testing.

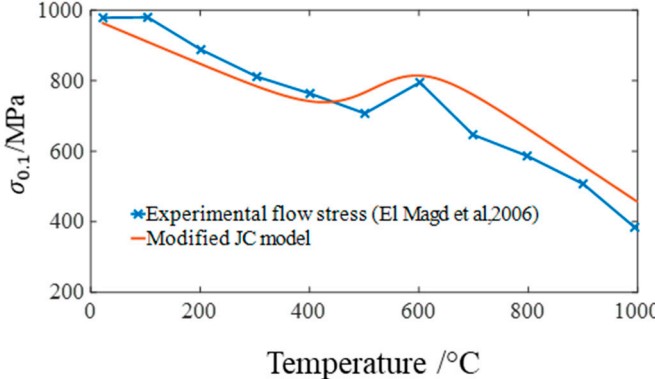

**Figure 4.** Experimental flow stress at $0.52 \times 104$ s$^{-1}$ from El Magd et al. [6] fitted using the MJC Model.

## 3. Flow Stress Modification due to Damage

JC and MJC models both assume no change in material behavior with increasing strain. However, in reality, the flow stress is altered because of fracture. Therefore, to model chip segmentation in ductile materials at a lower cutting velocity where the possibility of adiabatic shear is less incorporation of fracture is essential. Figure 5a provides the schematic modification of the flow stress curves due to fracture from the works of Childs et al. [16,26]. The flow stress before the fracture is defined by Equation (8) and is marked by 'BEFORE FRACTURE' in Figure 5a. During plastic deformation, at the microstructure level, nucleation and growth of defects, such as micro-voids & micro-cracks and their coalescence into macro-cracks takes place leading to material damage [27]. The damage due to plastic deformation in the workpiece is accumulated through the damage factor (*D*) and is defined, as shown in Equations (8)–(12) and Figure 5a.

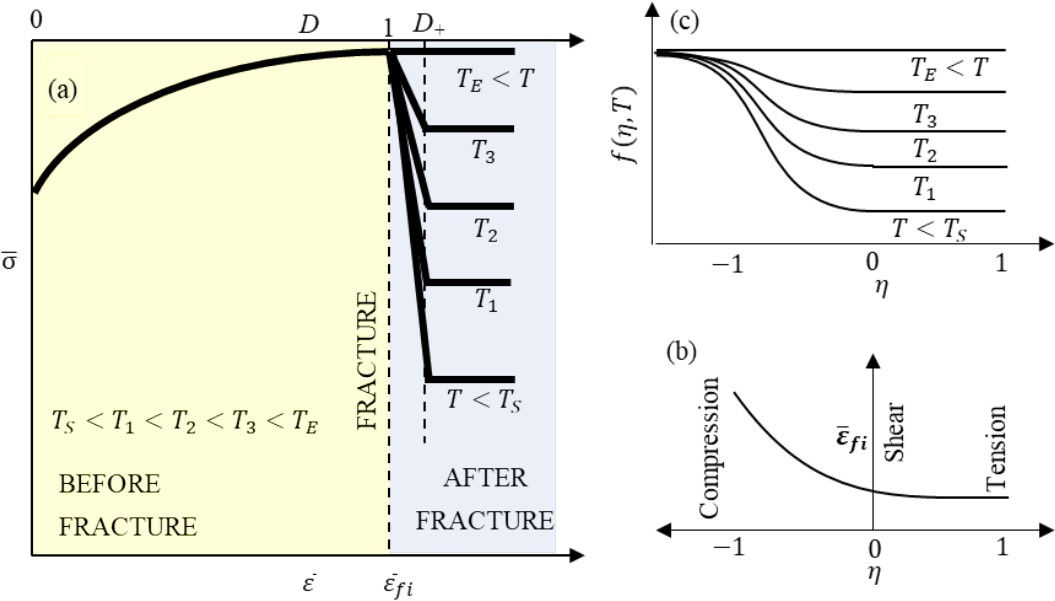

**Figure 5.** (**a**) Flow stress model for chip segmentation prediction with (**b**) flow stress modification factor as a function of stress triaxiality and temperature from Childs et al. using (**c**) the fracture initiation strain as a function of stress triaxiality.

The damage parameter is defined as the ratio of accumulated plastic strain to fracture initiation strain ($\varepsilon_{fi}$). The fracture initiation strain is presented in detail in the following section and it is the focus of the study. As the accumulated plastic strain equals fracture initiation strain, the damage parameter equals one and it is shown by "FRACTURE" in Figure 5a.

$$D = \int_0^{\overline{\varepsilon}} \frac{d\overline{\varepsilon}}{\overline{\varepsilon}_{fi}} \tag{8}$$

$$\overline{\sigma}_D = \begin{cases} \overline{\sigma} & D < 1 \\ q(\eta, T, D, D_+) \cdot \overline{\sigma} & 1 < D < D_+ \\ f(\eta, T) \cdot \overline{\sigma} & D_+ < D \end{cases} \tag{9}$$

$$q(\eta, T, D, D_+) = \left( \frac{1 + f(\eta, T)}{2} - \frac{1 - f(\eta, T)}{2} \tanh\left( a \frac{2D - D_+ - 1}{D_+ - 1} \right) \right) f(\eta, T) \tag{10}$$

$$f(\eta, T) = \begin{cases} tanh\left(-\sqrt{3}\mu_i\eta\right) & T < T_S \\ p(\eta, T) & T_S < T < T_E \\ 1 & T_E < T \end{cases} \tag{11}$$

$$p(\eta, T) = tanh\left(-\sqrt{3}\mu_i\eta\right) + \left[1 - tanh\left(-\sqrt{3}\mu_i\eta\right)\right]\left(\frac{T - T_S}{T_E - T_S}\right) \tag{12}$$

The modeling of the flow stress behavior after the fracture point is defined by "AFTER FRACTURE" in Figure 5a. The flow stress curve is modified based on the loading conditions, $\eta$, and temperature, $T$, as shown in Equation (8) using the flow stress modification factor, $f(\eta, T)$. In an earlier development of Childs et al. [16], $f(\eta, T)$ was assumed to be a constant. With further development [26], $f(\eta, T)$ is defined as a function of stress triaxiality and temperature, as shown in Equation (8) and Figure 5c and presented in detail in the following sections.

### 3.1. Fracture Initiation Strain

The need for the fracture initiation strain and its use in the damage factor is presented in the previous section. The fracture initiation strain is modeled as a function of stress triaxiality $(\eta)$, lode angle parameter $\left(\overline{\theta}\right)$, temperature $(T)$, and strain rate $\left(\dot{\varepsilon}\right)$, as shown in Equation (13).

$$\varepsilon_{fi} = f\left(\eta, \overline{\theta}\right) \cdot g\left(\dot{\varepsilon}\right) \cdot h(T) \tag{13}$$

The function, $f\left(\eta, \overline{\theta}\right)$, is used to define the loading conditions. The stress triaxiality $(\eta)$ defines the varying loading conditions in two-dimensional (2D) and the lode angle parameter, $\overline{\theta}$, extends the model to loading in three-dimensional (3D) space. In the case of the 2D cutting process (orthogonal cutting process), which falls under the plane strain condition, the lode angle parameter is zero. The stress triaxiality parameter defined by, $\eta$, as $\frac{\sigma_m}{\overline{\sigma}}$ defines the varying loading conditions from pure compression to combinations of shear/compression and shear/tension to tension, as shown in Figure 6.

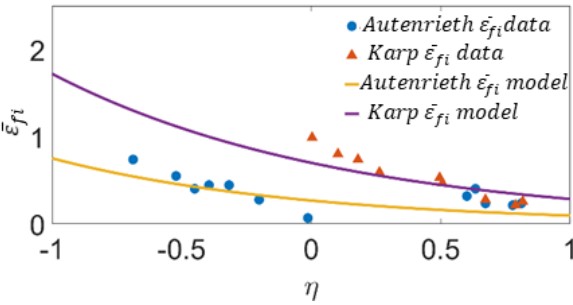

**Figure 6.** Fracture initiation strain predicted using Shear compression specimen under quasi-static loading, Shear specimens and compression testing as a function of stress triaxiality and fitted using an exponential function.

In order to obtain the fracture initiation strain under different conditions of $\eta$ (−1 to 1), specimens of varying geometrical shapes are to be used [28]. Interestingly, the fracture initiation strain for variants of C45E steel has been studied for nearly 75 years [29]. With the advances in material testing, the variants of the C45E steel's fracture strain has been studied. Autenrieth et al. [30] obtained the fracture strain as a function of stress triaxiality using torsion and torsion-tension tests. These tests provide stress triaxiality from zero to positive stress triaxiality conditions. Recently, the same fracture strain as a function of stress triaxiality for conditions that range from negative to positive has been obtained using the Shear-compression disk specimen by Karp et al. [31]. In this study, the latter two fracture strain models identified as Autenrieth fracture strain data and Karp fracture strain data are used to evaluate their influence on chip segmentation prediction. The difference in the fracture

strain between the two models can be attributed to the difference in processing history and the stress triaxiality evolution before fracture.

To implement the fracture strain in the finite element model, the exponential function used in the Bai–Wierzbicki damage model [32] is used and it is shown in Equation (14).

$$\varepsilon_{fi-X}(\eta) = D_{X1}e^{D_{X2}\eta} \tag{14}$$

The Autenrieth fracture strain data are fitted using the parameters ($D_{X1} = 0.2339$ and $D_{X2} = -1.035$) and the Karp fracture strain data are fitted using the parameters ($D_{X1} = 0.7$ and $D_{X2} = -0.9$).

The function $g(\dot{\varepsilon})$ is used to model the influence of strain rate. $h(T)$ is used to model the influence of temperature on the fracture initiation strain. In this study, $g(\dot{\varepsilon})$ and $h(T)$ are obtained from the JC damage fracture model, as shown in Equation (15) and Equation (16) and $D_3 = 0.0018$ and $D_4 = 0.58$ are obtained from the literature [33].

$$g(\dot{\varepsilon}) = \left(1 + D_3 \ln(\dot{\varepsilon}/\dot{\varepsilon}_0)\right) \tag{15}$$

$$h(T) = (1 + D_4 T^*) \tag{16}$$

### 3.2. Flow Stress Modification Factor

As previously mentioned, the flow stress is modified, as shown in Figure 5a using the flow stress modification factor, $f(\eta, T)$ shown in Figure 5c. The flow stress in the 'AFTER FRACTURE' region is modeled while using a steady-state flow stress modification factor defined by Equation (9) and a transient flow stress modification factor. The steady-state is defined by $D > D+$ and the transient state is defined by $D+ > D > 1$. In this study, $D+$ is defined to be 1.25, as suggested by Childs et al. [26].

In the steady-state, the flow stress modification is a function of stress triaxiality and it is defined by Equation (12) below a certain temperature, $T_S$, as shown in Figure 5c. A negative stress triaxiality condition (e.g., $\eta = -1$) characterizing compression, the flow stress is modified only slightly due to incompressibility. A positive stress triaxiality condition, e.g., $\eta = 1$ characterizing tension, the flow stress is drastically reduced to zero, similar to necking in tensile testing. At temperatures above $T_E$, the material is assumed to be healed and continue to flow plastically with the flow stress defined by $\bar{\sigma}$, as shown in Equation (8). Between temperatures $T_S$ and $T_E$, the flow stress curve is defined using the tanh function, as shown in Equation (11). The transient flow stress modification factor is defined as Equation (9) by multiplying the steady-state flow stress modification factor with a tanh function of $D$ and $f(\eta, T)$.

The values for $\mu_i$, $T_S$, and $T_E$ used in this work are 1 °C, 600 °C, and 700 °C, respectively. The values for $T_S$ and $T_E$ based on the strain hardening transition behavior observed from the Gleeble tests.

## 4. Experimental Investigation of Chip Formation in Orthogonal Turning

The orthogonal turning process has been carried out in a normalized C45E steel tube with a diameter of 150 mm with a thickness of 3 mm, determining the uncut chip width. The material hardness is measured and an average hardness of 220 HV is recorded with an average grain size of 13.5 µm. The tubular workpiece reduces grain size variation influence. All of the experiments were carried out under dry cutting conditions. The cutting tool material is H13A grade carbide with TiCN coating. The cutting velocity is set at 150 m/min. The rake angle is varied as −5°, 5°, 10°, and 20°, and the feed is varied as 0.05 mm.rev$^{-1}$, 0.1 mm.rev$^{-1}$, 0.15 mm.rev$^{-1}$, 0.25 mm.rev$^{-1}$, 0.4 mm.rev$^{-1}$, and 0.6 mm.rev$^{-1}$. The cutting forces and feed forces are obtained through the Kistler dynamometer attached to the tool turret. The sample chips were collected for all cutting conditions and they have been used to quantify chip segmentation, as shown in Figure 7. The experimental investigation is described in detail in previous work [34]. The chips have to be plotted with varying magnification to capture the chip segmentation features at very low feed rate conditions (e.g., 0.05 mm.rev$^{-1}$) and similarly at very high

feed rate conditions (e.g., 0.60 mm.rev$^{-1}$). This approach with varying magnification is carried out when the chips from simulations are also presented in the following sections.

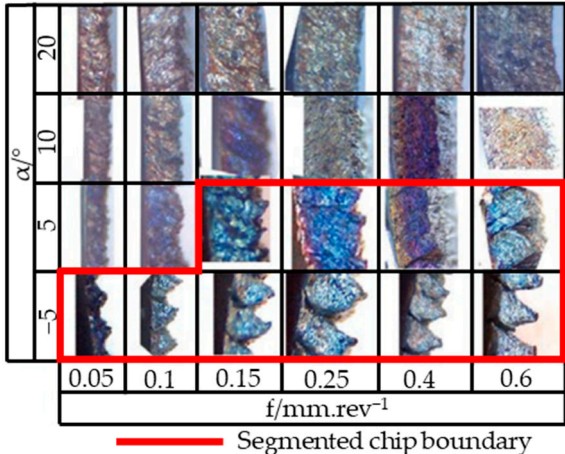

**Figure 7.** Chip morphology obtained through experimental investigation under varying rake angle and feed at a constant cutting speed of 150 m/min. with the boundary differentiating between continuous and segmented chip. Note: Image magnification not to scale.

## 5. FE Modeling of Chip Formation in Orthogonal Turning Process

The FE simulation of the cutting process was carried out using a commercial finite element software, Thirdwave Advantedge [35]. The software has been specifically built ground up for metal cutting simulations while using a dynamic explicit coupled Thermo-elastoplastic Lagrangian formulation [36].

The main requirement for metal cutting simulation is the ability for adaptive remeshing to account for the large plastic strains in the primary and secondary deformation zones. The simulations are carried out in 2D. The software provides the ability to customize the adaptive remeshing parameters. The adaptive remeshing parameters control the mesh transition in the tool vicinity (primary deformation zone) from coarse to fine and also the mesh transition from fine to coarse (chip mesh). The software through the mesh refinement and coarsening factors automatically control the conditions for adaptive remeshing. With predicting chip segmentation being the main aim of the study, mesh refinement factor of 6 (1-coarse → 8-fine) and mesh coarsening factor of 3 (1-fine → 8-coarse) is chosen.

A further increase in mesh refinement factors leads to computation time to increase by a factor of 3, making it practically unviable. Besides, the suggested minimum element size is set at 0.01 mm. The friction coefficient of 0.5 is used for all cutting conditions. The software provides the possibility to input the custom material model through a FORTRAN Subroutine, which and has been employed in the study to implement the modified JC model. Two different solution algorithms are supported by the software: Newton method and the secant method. At this stage, the secant method is used, as the method does not require the derivative implementation with the compromise of a slower convergence rate.

Simulations are carried out for all rake angles (−5°, 5°, 10° and 20°) and feed from 0.05 mm.rev$^{-1}$ to 0.6 mm.rev$^{-1}$. The relief angle and the cutting-edge radius are kept constant with the values 7° and 30 μm, respectively, both in the experimental conditions and simulations. Simulations are run with the JC model and MJC model combined with the damage modeling approach. Within the MJC model + damage model framework, the influence of two fracture initiation strain model's (Autenrieth $\varepsilon_{fi}$ model and Karp $\varepsilon_{fi}$ model) capability to evaluate the transition from continuous chip to segmented chip is studied.

## 6. Results

The FE simulations were carried out using two different material models (JC and MJC). With the MJC+ damage model, two different parameter sets of fracture initiation strain models' ability to predict the continuous chip to segmented chip transition are evaluated. The FE simulations were run till a steady state was achieved, and the chip morphology was obtained. The chip morphology was plotted in the chip chart form. Further on, the continuous chip–segmented chip boundary is identified in the chip chart. The cutting forces predicted by the material models are also presented.

### 6.1. Validation of Material Model under Continuous Chip Formation Conditions

This section presents the MJC model validation in the simulation of the chip formation process. Experimental results with continuous chips are used to avoid the damage model influence. The rake angle of 10° was used with feeds of 0.05 mm.rev$^{-1}$ to 0.6 mm.rev$^{-1}$ and Figure 8 presents the cutting forces.

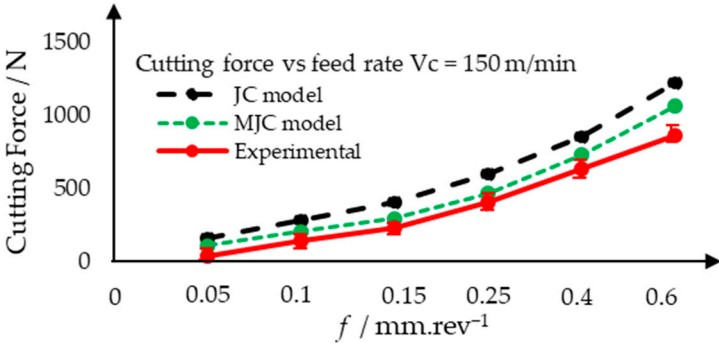

**Figure 8.** Cutting force predicted by the JC model and modified the JC model for a constant cutting speed of 150 m/min and rake angle of 10°.

The experimental result showing the increase in cutting force with feed is well established within the metal cutting with the influence of uncut chip thickness increase. The cutting forces predicted by the JC model and the MJC model have a similar trend. The MJC model improves the prediction for all of the cutting conditions quantitatively. This improvement in cutting force prediction as compared to the JC model is attributed to the material testing used and the ability of the MJC model to incorporate the material behavior with improved accuracy. The highest discrepancies in quantitative prediction for both JC and MJC models are at the maximum chip thickness conditions. The MJC model's cutting force prediction error can be attributed to the extrapolation that was carried out with the fitting of the MJC model at very high strain rates. Further improvement has to be carried out, among other things, by improving the friction models.

### 6.2. Experimental and FE Modeling of Continuous Chip–Segmented Chip Transition for Varying Rake Angle and Feed

The chips obtained from experimental investigation for varying rake angle and feed rates at a constant cutting speed of 150 m/min are plotted in the form of chip chart in to evaluate the ability of the different fracture initiation strain models in chip segmentation prediction. Figure 7 shows that, at constant cutting speed, continuous chips are produced for rake angles 10° and 20°. With a rake angle of 5°, the cutting process is in a transition zone. Continuous chips are produced for feed of 0.05 mm.rev$^{-1}$ and 0.10 mm.rev$^{-1}$ and segmented chips are produced for feed from 0.15 mm.rev$^{-1}$ to 0.6 mm.rev$^{-1}$. For a rake angle of −5°, the produced chips are segmented for all feeds. At a constant feed of 0.40 mm.rev$^{-1}$, a change in the rake angle from 5° to 10° changes the chips from being continuous to segmented.

Similarly, at a constant rake angle of 5°, the change in feed rate from 0.10 mm.rev$^{-1}$ to 0.15 mm.rev$^{-1}$ leads to continuous to segmented chip. This leads to the rake angle of 5° being a transition zone. The experimental results clearly show that chip segmentation/continuous chip is stable under certain cutting conditions (−5°, 10°, and 20°) and in transition mode under certain conditions (5°). It is also seen that, in the transition zone, the feed also plays a role in chip segmentation. This states that minor changes in the cutting zone to be highly sensitive to the machining output. This would lead to challenges in the process capability in a production environment.

*6.3. Prediction of Chip Segmentation Using MJC Model and Two Different Fracture Initiation Strain Models*

Initial simulations were run with the JC flow stress model [37] and the JC fracture model [38] reported for AISI 1045 steel to predict chip segmentation. The continuous chip to segmented chip transition predicted by the JC flow stress-JC fracture model, as shown in Figure 9, with the JC fracture parameters that were obtained from the literature [38]. The JC flow stress–JC fracture model is unable to predict the influence of the rake angle and, at the same time, predicts lower chip segmentation intensity when compared to experimental investigation. With the inability of the JC flow stress–JC fracture model established, the flow stress behavior is modified while using the damage model to improve chip segmentation prediction. The chip morphology predicted by the MJC model with the damage model using the two fracture initiation strain models is presented here.

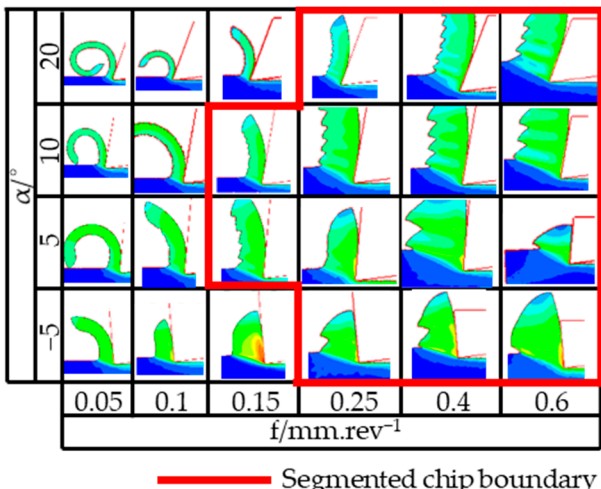

**Figure 9.** Continuous chip to segmented chip transition prediction predicted by the JC flow stress model in combination with the JC Fracture model. Note: Image magnification not to scale.

*6.4. Autenrieth Fracture Initiation Strain Model Predicted Chip Segmentation Continuous Chip Transition*

Figure 10 shows the transition of the chip morphology that was predicted by the Autenrieth fracture initiation strain model and MJC material model. The continuous chip to segmented chip transition is predicted with reasonable accuracy. At a constant feed of 0.6 mm.rev$^{-1}$, the model can predict the segmented chip for rake angles: −5° and 5° and the transition to continuous chip with a rake angle of 10°. The model can predict the transition from continuous chip to segmented chip as the feed is increased from 0.05 mm.rev$^{-1}$ to 0.1 mm.rev$^{-1}$. At a feed of 0.05 mm.rev$^{-1}$, a continuous chip is predicted for both negative and positive rake angle. A qualitative comparison of chip segmentation between the experimental investigation and the Autenrieth fracture initiation strain model prediction shows that the level of chip segmentation intensity is lower when compared to the experimental results. This leads to a further quantitative evaluation in terms of chip segmentation intensity become counterproductive. Chip segmentation intensity has been predicted with improved accuracy in literature but only for a constant rake angle and varying cutting speeds [39]. The models in these studies have shown cutting speed's influence and not the rake angle's influence on chip segmentation.

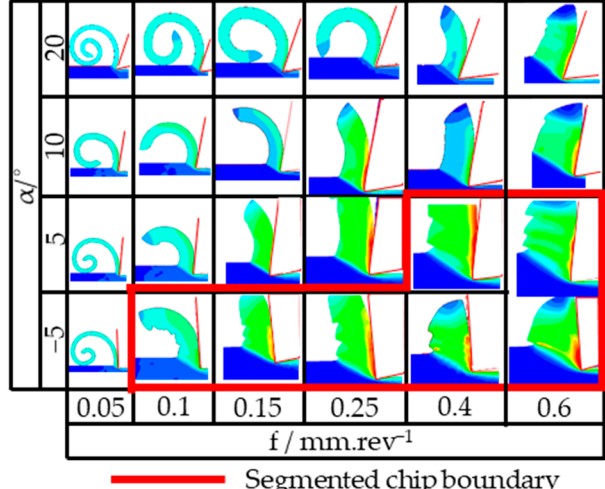

**Figure 10.** Continuous chip to segmented chip transition prediction predicted by the MJC flow stress model in combination with the Autenrieth Fracture initiation strain model. Note: Image magnification not to scale.

*6.5. Fracture Strain Through Shear Compression Disk (SCD) Experiments Predicted Chip Segmentation Continuous Chip Transition*

Figure 11 shows the transition of the chip morphology predicted by the fracture initiation strain that was obtained by Karp et al. [31] while using shear compression disk specimen. The model predicts segmented chips for all rake angles for feeds above and including 0.1 mm.rev$^{-1}$. For the feed of 0.05 mm.rev$^{-1}$, the segmented chip for a negative rake angle is not predicted correctly compared to experimental results. Qualitative evaluation of the chip segmentation intensity when compared to the experimental investigation is relatively low for a negative rake angle. On the other hand, the chip segmentation frequency is higher qualitatively compared to the Autenrieth fracture initiation strain model predicted chip segmentation frequency.

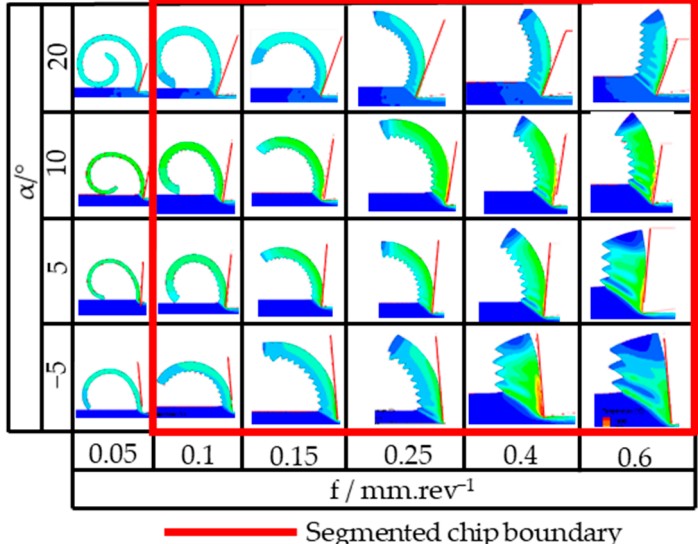

**Figure 11.** Continuous chip to segmented chip transition prediction by the fracture initiation strain obtained Shear compression specimen. Note: Image magnification not to scale. Note: Image magnification not to scale.

## 7. Discussion

The FE simulation of the machining process's ability to predict chip segmentation is dependent on the accuracy in which the material behavior is modeled to a large extent, as shown by Fernandez-Zeliaia et al. [40]. The other well-known influencing factors are friction behavior and thermal behavior input. Within flow stress behavior, the initial yield stress, the strain hardening behavior, the fracture initiation, and the material behavior after fracture are essential inputs. For the material under study, C45E steel, strain hardening behavior, and thermal softening behavior are more influential when compared to strain rate hardening [41]. The MJC model's ability to accurately predict three different strain hardening behavior as a function of temperature, therefore, systematically improves the accuracy of FE modeling of chip formation. The material behavior at the strain, strain rate, and temperature occurring in the chip formation conditions is vital in accurately predicting the chip morphology. In this study, the material flow stress is initially validated while using the flow stress that was obtained from compression testing through Gleeble tests (Figure 3). Although the temperatures at which the material behavior is validated through Gleeble tests are comparable, the strains and strain rates are lower by orders of magnitude. The material model validation through cutting process simulation is carried out by avoiding the cutting conditions where chip segmentation is not observed. The MJC model's DSA incorporation, which is active in temperatures occurring at the primary and secondary deformation zones, is an important outcome of this work. The ability to incorporate DSA keeping all other things constant has been shown to improve the cutting forces prediction accuracy, as shown in Figure 8. The fracture initiation models with the JC model have not been able to predict the chip segmentation boundary. This shows the influence of the MJC model on the improved prediction of chip segmentation boundary.

The damage behavior modeling involves fracture initiation strain and the flow stress after the fracture initiation strain. The results of Figure 9, Figure 10, and Figure 11 show that the continuous chip to segmented chip transition and chip segmentation intensity is sensitive to the fracture initiation strain input. The Autenrieth fracture initiation strain model better predicts the transition between the continuous chip and segmented chip when compared to the Karp fracture initiation strain model. Simulations run with the JC material model combined with the fracture initiation strain models have shown poor results in terms of prediction of the chip segmentation (Figure 9). This shows that the ability of the fracture initiation strain models to predict chip segmentation is dependent on the components of the flow stress behavior, i.e., initial yield stress and strain hardening behavior. The strain hardening behavior and the resulting temperature increase influences the fracture initiation strain. The strain rate difference and the initial yield stress variation between the two experimental conditions (for the Autenrieth fracture initiation strain model and Karp fracture initiation strain model) have been reported as being possible causes for the significant variation in the fracture initiation strain. The significant fracture initiation strain variation correspondingly leads to significant variation in the continuous chip to segmented chip transition and the chip segmentation intensity prediction in the finite element models that were developed in this study.

The experimental investigation (Figure 7) shows that the chip segmentation at a constant cutting speed is influenced by tool geometry and chip thickness. In this study, the normal rake angle is the tool geometry parameter modified, and its influence on the chip formation is studied. With the normal rake angle being modified from negative to positive rake angle under certain uncut chip thickness (feed) conditions, the chip formation process is transformed from a continuous chip to a segmented chip, as observed with the segmented chip boundary.

With the normal rake angle modification, the mechanical loading of the workpiece ahead of the tool is modified with varying levels of shear and compression. The mechanical loading is not uniform along the shear plane from the cutting edge to the free surface in the primary deformation zone. The stress that was observed by the workpiece material close to the cutting-edge is highly compressive when compared to the free end of the primary deformation zone, where the stresses are a combination of shear and compression. The workpiece material that is close to the cutting-edge rounding is mainly

under the stress state defined by combined shear and compression defined by a large negative stress triaxiality. The workpiece material on the free end of the primary deformation zone has a relatively lower negative stress triaxiality; still, they are in the negative stress triaxiality conditions. Figure 6 shows that the fracture initiation strain in the negative stress triaxiality condition to be sensitive and varies based on testing conditions. This fracture initiation strain sensitivity is attributed to be the main reason for chip segmentation to be sensitive to minor changes in cutting conditions. At a constant chip thickness and cutting speed, the tool geometry influences this mechanical loading and it is observed to influence the onset of chip segmentation. Another critical factor that influences the material behavior is the material's history. From a machining perspective, the influence of the previous cut on the strain profile has been shown by Childs et al. [26]. The tool edge geometry controls the influence of the previous cut on the stress profile on the workpiece surface, tool wear, and tool vibration. The tool edge geometry could also influence the transition zone between the continuous chip to segmented chip. In this work, the influence of the previous cut is not taken into account and it could be a source of improvement in chip segmentation prediction in future work.

It is clearly understood that the uncut chip thickness, in addition to the rake angle, influences the transition from continuous to segmented chip at constant cutting speed. This is understood from the results, showing that, at a constant rake angle, the chip thickness influences chip segmentation. The chip thickness increase leads to the heat being concentrated in the primary shear zone. This concentration of heat leads to a segmented chip, as predicted in the presented FE simulations. On the other hand, chip thickness increase needs to be significant in this case at a rake angle of $5°$ up to $0.6$ mm.rev$^{-1}$ to produce segmented chips.

## 8. Conclusions

In this study, a new modified JC model is developed based on the Voyiadjis–Abed–Rusinek [8] constitutive modeling approach. The modified JC model with the fracture modeling approach by Childs et al. [26] has been implemented for the first time in the FE framework for the chip formation simulation. The influence of the fracture initiation strain model on the continuous to segmented chip transition prediction has been successfully demonstrated. The following conclusions are made from the study.

1.  The modified JC model can incorporate the DSA influence on flow stress curves of normalized C45E steel for varying temperatures and strain rates with reasonable accuracy.
2.  The strain hardening behavior modeling of the modified JC model can predict the temperature's influence on strain hardening accurately when compared to the JC model.
3.  The strong influence of DSA in the simulation of chip formation in machining of C45E steel is established with an improvement of cutting force prediction accuracy.
4.  The transition from continuous chip to segmented chip is established to be a function of normal rake angle and feed at a constant cutting speed in orthogonal cutting.
5.  The Autenrieth fracture initiation strain model can predict the chip segmentation boundary better as compared to the Karp fracture initiation strain model.

**Author Contributions:** Conceptualization, A.M.D.; Formal analysis, A.M.D.; Supervision, P.V.S., T.B. and M.E.; Validation, A.M.D. and P.V.S.; Writing–original draft, A.M.D.; Writing–review & editing, P.V.S., T.B. and M.E. All authors have read and agreed to the published version of the manuscript.

**Funding:** This research was funded by Sandvik Coromant AB and the Knowledge Foundation through the Industrial Research School SiCoMaP, Dnr 20110263, 20140130.

**Acknowledgments:** The authors would like to thank T.H.C. Childs for his support and guidance with the finite element implementation of the damage model and for invaluable discussions.

**Conflicts of Interest:** The authors declare no conflict of interest.

## Nomenclature

| Symbol | Description | Unit |
|---|---|---|
| $A$ | Initial yield stress | MPa |
| $A_0$ | Initial yield stress at room temperature and reference strain rate | MPa |
| $A_\Delta$ | Flow stress increase function in DSA range in modified JC model | MPa |
| $a$ | Damage function parameter (Childs damage function) | - |
| $B$ | Strain hardening coefficient | MPa |
| C | Strain rate hardening coefficient | - |
| $D$ | Damage | - |
| $D_+$ | Steady-state damage value | - |
| $D_{Xi}$ | Fracture initiation parameter | - |
| $D_3$ | Strain rate component coefficient of fracture initiation function | - |
| $D_4$ | Temperature component coefficient of fracture initiation function | - |
| f | Feed rate | mm.rev$^{-1}$ |
| $T_l$ | Lower temperature bound function in DSA range (MJC Model) | °C |
| $T_{ll}$ | DSA range's lower temperature bound at room temperature and reference strain rate (MJC Model) | °C |
| $T_h$ | Upper-temperature bound function of DSA range | °C |
| $T_{hl}$ | Upper temperature bound of DSA range at room temperature and reference strain rate (MJC Model) | °C |
| $T$ | Temperature | °C |
| $T_o$ | Reference temperature | °C |
| $T_m$ | Melting temperature | °C |
| $T_S$ | Lower bound healing temperature (Childs damage function) | °C |
| $T_E$ | Upper bound healing temperature (Childs damage function) | °C |
| $T^*$ | Homologous temperature | - |
| $t$ | DSA regime fitting parameter of MJC Model | |
| m | Thermal softening coefficient | - |
| n | Strain hardening exponent | - |
| $\alpha$ | Rake angle | ° |
| $\xi$ | Temperature dimensioned parameter of the VAR model | - |
| $\Delta\sigma$ | Flow stress increase of DSA range in MJC model at room temperature and reference strain rate | MPa |
| $\overline{\varepsilon}_{fi}$ | Fracture initiation strain | - |
| $\overline{\varepsilon}$ | Equivalent plastic strain | - |
| $\varepsilon_{fi-X}$ | Fracture initiation strain | - |
| $\dot{\varepsilon}$ | Strain rate | s$^{-1}$ |
| $\dot{\varepsilon}_0$ | Reference strain rate | s$^{-1}$ |
| $\eta$ | Stress triaxiality | - |
| $\overline{\theta}$ | Lode angle parameter | - |
| $\mu_i$ | Internal Coulomb friction coefficient (Childs damage function) | - |
| $\overline{\sigma}$ | Flow stress | MPa |

**Abbreviation**

| | | |
|---|---|---|
| FE | Finite element | |
| DSA | Dynamic strain aging | |
| VAR | Voyiadjis–Abed–Rusinek model | |
| JC | Johnson-Cook model | |
| MJC | Modified Johnson-Cook model | |
| SCD | Shear compression disk specimen | |

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
