# Peer review of "Predicting Continuous Chip to Segmented Chip Transition in Orthogonal Cutting of C45E Steel through Damage Modeling"

_metals, doi:10.3390/met10040519_

Round 1

Reviewer 1 Report

This paper presents an important and up-to-date problem concerning the prediction of chip shape transitions during machining. However in order to accept, some revision is required. The detailed remarks are as follows:

  • Paper contains many symbols, thus it is advised to include to nomenclature list.
  • The sentence in abstract: "The study shows for the first time predicts the continuous chip successfully to segmented chip transition for varying feed and rake angle" is incomprehensible. Please correct it.
  • Despite the torough state-of-the-art, some recent researches concerning the FEM modeling of machining processes are omitted. Therefore, please include the discussion on following works in a literature survey:
  • 1. Evaluation of Ductile Fracture Models in Finite Element Simulation of Metal Cutting Processes, J. Manuf. Sci. Eng. 136 (2014). doi: 10.1115/1.4025625.
  • 2. An evaluation of different damage models when simulating the cutting process using FEM, Procedia CIRP, 58 (2017) 134-139. https://doi.org/10.1016/j.procir.2017.03.202.
  • 3. Evaluation of chip formation simulation models for material separation in the presence of damage models, Simul. Model. Pract. Th. 19 (2011) 718-733. https://doi.org/10.1016/j.simpat.2010.09.006.
  • 4. Prediction of cutting forces during micro end milling considering chip thickness accumulation, International Journal of Machine Tools and Manufacture (2019), 103466, https://doi.org/10.1016/j.ijmachtools.2019.103466.
  • Page 2, line 84: the reference source is not found - please correct it.
  • Please put the numbering to the bolded parts of manuscript, e.g. "Modified Johnson-Cook model:"
  • There is no reference to Fig. 1 in the text of manuscript - please correct it.
  • The figures 7, 9, 10, 11 are difficult to read. Please improve their quality.
  • Please try to explain in more details, why the measured force values are underestimated comparing to the predicted ones.
  • Please check once again language in paper. It contains some typo errors.

Author Response

Dear Respected Reviewer,

On the outset, we would like to thank you for your critical and specific inputs in improving the quality of the submitted article. We have taken each input with utmost importance and improved the article to our capability. We would also like to thank you for your general positive comment on our submission

 We have answered each to each of your comments and suggestions.

1. Paper contains many symbols, thus it is advised to include to nomenclature list.

Answer: We have added a list of symbols and the abbreviations in a nomenclature list based on your input at the end of the article.

2. The sentence in abstract: "The study shows for the first time predicts the continuous chip successfully to segmented chip transition for varying feed and rake angle" is incomprehensible. Please correct it.

Answer: We have modified the sentence and make it comprehensible as follows. "The study shows the transition from continuous chip to segmented chip for varying feed rates and rake angles for the first time. "

3. Despite the torough state-of-the-art, some recent researches concerning the FEM modeling of machining processes are omitted. Therefore, please include the discussion on following works in a literature survey:

Answer: We have incorporated the literatures as mentioned.

4. Page 2, line 84: the reference source is not found - please correct it.

Answer: The mentioned reference is to the equation number. The manuscript has been revised and rewritten as Equation (2) instead of (2) for more clarity.

5. Please put the numbering to the bolded parts of manuscript, e.g. "Modified Johnson-Cook model:"

Answer: We have corrected the manuscript using the MDPI styles present in the template to correct the mentioned error.

6. There is no reference to Fig. 1 in the text of manuscript - please correct it.

Answer: We have updated the reference to Figure 1 and have cross checked for all figures and equations.

7. The figures 7, 9, 10, 11 are difficult to read. Please improve their quality.

Answer: Thank you for the input. We have increased the figure size and the font size according to improve the quality. 

8. Please try to explain in more details, why the measured force values are underestimated comparing to the predicted ones.

Answer: Thank you for the comment. We have added additional arguments for the possible differences in terms of force prediction of the proposed models.

9. Please check once again language in paper. It contains some typo errors.

Answer: Thank you for the comment. We have revised the manuscript to improve the readability of the text and have also used professional grammar check for improving the general quality.

We would like to thank the reviewer again for the positive constructive comments for the improvement of the submission.

Yours sincerely

Ashwin Devotta

Corresponding author.

Reviewer 2 Report

The article can be proposed for publication after revision. The following recommendations should be considered when revising:

  1. The text of the article must be thoroughly checked for the presence and correctness of links. A message about the absence of a source appears several times in the text (“reference not found”, s. p. 2, line 84, p. 5, line 130, h. 7, line 178-179). The reference should be given to previously reported work (s. p. 3, line 96). In the reference to the source [35] in the text (s. p. 13, line 330), it is necessary to indicate the surname (“Fernandez-Zeliaia”), not the first name (“Patxi”).
  2. All abbreviations used must be decrypted in the text (see, for example, MJC and SCD).
  3. All figures must be given links in the text, as well as to verify the correctness of links to graphic material (For example, there is no link to figures 1, 2, 4).
  4. The equations must be numbered consecutively (The numbers of equations 2a, 3b, etc. must be replaced by 1, 2, 3, ...). All equations must be referenced in the text (For example, there is no link to equation 2 and equation 4). It is necessary to check the correctness of references to equations (For example, the equations on page 2 and page 3 have the same number "1").
  5. All graphs should be brought to the same style: names and dimension of axes, the same font of all graphs, etc.
  6. The equations shown in the figures should be fully visible (see, for example, figures 1, 4). Axes and graphs should not cross the text (see for example figure 5). In figures 7 and 11, it is necessary to provide a larger cell size, since the text intersects with the cell lines. The cells of figures 9 and 10 must be filled in with the corresponding numbers and text.
  7. All figures with the image of chips, both experimental and obtained as a result of modeling, must be performed on the same scale.
  8. When describing the conditions for experimental studies, it is necessary to indicate the hardness of the processed material and the step of the feed change.
  9. It is necessary to indicate at what values of deformations the remeshing was performed.
  10. In the section “Validation of material model …” (s. p. 10), it is necessary to indicate the dispersions of the forces experimental values and give the deviations of the calculated and experimental values.

Author Response

Dear Respected Sir,

On the outset, we would like to thank you for your critical and specific inputs in improving the quality of the submitted article. We have taken each input with utmost importance and improved the article to our capability. We would also like to thank you for your general positive comment on our submission

 We have answered to each of your comments and suggestions.

1. The text of the article must be thoroughly checked for the presence and correctness of links. A message about the absence of a source appears several times in the text (“reference not found”, s. p. 2, line 84, p. 5, line 130, h. 7, line 178-179). The reference should be given to previously reported work (s. p. 3, line 96). In the reference to the source [35] in the text (s. p. 13, line 330), it is necessary to indicate the surname (“Fernandez-Zeliaia”), not the first name (“Patxi”).

Answer: Thank you for your detailed comments. We have incorporated the changes that you have mentioned. The difference between cross referencing of equations and article reference has been highlighted by using the caption "Equation". 

2. All abbreviations used must be decrypted in the text (see, for example, MJC and SCD).

Answer: By the input provided by the reviewer, all the abbreviations have been decrypted in the text.

3. All figures must be given links in the text, as well as to verify the correctness of links to graphic material (For example, there is no link to figures 1, 2, 4).

Answer: With the input provided by the reviewer, the cross referencing of the graphic material has been incorporated accordingly.

4. The equations must be numbered consecutively (The numbers of equations 2a, 3b, etc. must be replaced by 1, 2, 3, ...). All equations must be referenced in the text (For example, there is no link to equation 2 and equation 4). It is necessary to check the correctness of references to equations (For example, the equations on page 2 and page 3 have the same number "1").

Answer: With the input provided by the reviewer, the equation numbers have been changed and accommodated in the revised manuscript.

5. All graphs should be brought to the same style: names and dimension of axes, the same font of all graphs, etc.

Answer: The graphs have been updated based on the reviewers inputs and the quality of the graphs have been enhanced.

6. The equations shown in the figures should be fully visible (see, for example, figures 1, 4). Axes and graphs should not cross the text (see for example figure 5). In figures 7 and 11, it is necessary to provide a larger cell size, since the text intersects with the cell lines. The cells of figures 9 and 10 must be filled in with the corresponding numbers and text.

Answer: The figures 1, 4, 5, 7 and 11 have been corrected based on the reviewer's inputs and the quality has been enhanced.

7. All figures with the image of chips, both experimental and obtained as a result of modeling, must be performed on the same scale.

Answer: Thank you for the input provided. Unfortunately, we are unable to draw the figures in the same scale. when the chips are plotted in the same scale, the features of interest, (chip segmentation) is not visible for the low feed rates of 0.05 mm.rev-1, 0.1 mm.rev-1 and 0.15 mm.rev-1. For clarity, we have incorporated ther reason for different scale is mentioned in the manuscript.

8. When describing the conditions for experimental studies, it is necessary to indicate the hardness of the processed material and the step of the feed change.

Answer: Thank you for the input provided. We have added the information requested in the manuscript pg. 8 line number: 223-224.

9. It is necessary to indicate at what values of deformations the remeshing was performed.

Answer: Thank you for the improvement. We do not explicitly specify the values of deformations for remeshing due to the software we have used. On the other hand, the parameters for level of remeshing has been mentioned in the manuscript in page 9 line 247-248.

10. In the section “Validation of material model …” (s. p. 10), it is necessary to indicate the dispersions of the forces experimental values and give the deviations of the calculated and experimental values.

Answer: The values of the dispersions has been incorporated based on the reviewer's input in Figure 8.

We would like to thank the reviewer again for the positive constructive comments for the improvement of the submission.

Yours sincerely

Ashwin Devotta

Corresponding author.

Round 2

Reviewer 2 Report

Dear authors,
Thank you for the changes in your manuscript according to my comments.
I think that the article can be recommended for publication in this form.